# Negatively charged nanoporous membrane for a dendrite-free alkaline zinc-based flow battery with long cycle life

Zhizhang Yuan[1], Xiaoqi Liu[1,2], Wenbin Xu[1], Yinqi Duan[1], Huamin Zhang[1,3] & Xianfeng Li [1,3]

Alkaline zinc-based flow batteries are regarded to be among the best choices for electric energy storage. Nevertheless, application is challenged by the issue of zinc dendrite/accumulation. Here, we report a negatively charged nanoporous membrane for a dendrite-free alkaline zinc-based flow battery with long cycle life. Free of zinc dendrite/accumulation, stable performance is afforded for ~240 cycles at current densities ranging from 80 to 160 mA cm$^{-2}$ using the negatively charged nanoporous membrane. Furthermore, 8 h and 7 h plating/stripping processes at 40 mA cm$^{-2}$ yield an average energy efficiency of 91.92% and an areal discharge capacity above 130 mAh cm$^{-2}$. A peak power density of 1056 mW cm$^{-2}$ is achieved at 1040 mA cm$^{-2}$. This study may provide an effective way to address the issue of zinc dendrite/accumulation for zinc-based batteries and accelerate the advancement of these batteries.

[1] Division of Energy Storage, Dalian Institute of Chemical Physics, Chinese Academy of Sciences, 457 Zhongshan Road, Dalian 116023, P. R. China. [2] University of Chinese Academy of Sciences, Beijing 100049, P. R. China. [3] Collaborative Innovation Center of Chemistry for Energy Materials (iChEM), Dalian 116023, P. R. China. These authors contributed equally: Zhizhang Yuan, Xiaoqi Liu and Wenbin Xu. Correspondence and requests for materials should be addressed to X.L. (email: lixianfeng@dicp.ac.cn)

Pressing global environmental concerns and declining fossil energy sources have promoted an urgent need for energy storage technologies that can be coupled with renewable energies, such as wind and solar power[1,2]. Flow batteries have been regarded as one of the most promising technologies for large-scale energy storage due to their attractive features of high safety, high efficiency, and long cycle life[3]. The vanadium flow battery is one of the most promising technologies, which is at the stage of the commercial demonstration. However, the relatively high cost and low energy density limit extensive commercial application in large-scale energy storage. Nonaqueous flow batteries possess attractive features of wide electrochemical windows and broad selection of redox-active materials. However, the poor ionic conductivity of organic-based electrolytes imparts a low current density, and further leads to low-power density[4]. In contrast, aqueous flow batteries are a promising option due to the high-power density and safety.

Among the reported aqueous flow batteries, zinc-based flow batteries and alkaline zinc–iron-based flow batteries, in particular, have triggered attention due to their attractive features of high open-cell voltage (low electrochemical potential of zinc redox couple in alkaline electrolyte, −1.245 V vs. SHE)[5,6], low cost, and environmental friendliness. An alkaline zinc ferricyanide flow battery was first reported by G.B. Adams et al. in 1981[7]. Afterward, further work on this type of flow battery has tapered off. Currently, although ViZn Energy is developing an alkaline zinc–iron flow battery for grid-scale application[1], progress has been rarely reported mainly due to their short cycle life induced by zinc dendrite/accumulation, which is a common issue for zinc-based flow batteries[6,8,9]. The zinc dendrite/accumulation becomes even more serious at higher working current density, thereby limiting the power density of the battery. Therefore, considerable efforts have been devoted to address the issue of zinc dendrite/accumulation in zinc-based batteries, especially in alkaline medium, for which introducing additives such as adding ethanol (EtOH), $Pb_3O_4$, and $Na_2WO_4$ in the electrolytes is well-known[9,10]. Unfortunately, the additives normally result in a high polarization of the electrode, further leading to diminished battery performance. Another effective way of addressing the zinc dendrite has been realized through a backside-plating configuration that inhibits short circuits from zinc metal dendrites in the anode[5]. However, this backside-plating configuration brings a twofold increase in solution resistance over the frontside-plating configuration.

Here, we design a nanoporous membrane with negative charge (Fig. 1) on the pore walls and surface, which can tackle the zinc dendrite/accumulation issue and thus afford the battery with a long cycle life. In this design, the plating of zincate ions can be easily induced from the membrane direction (Fig. 1) to the 3D carbon felt framework direction (Fig. 1) through the mutual repulsion between the negatively charged zincate ions and the negatively charged surface and pore walls of the nanoporous membrane. Thus, even if zinc dendrites form, they grow through the backward direction of the membrane, which prevents the membrane from being broken up and stops the battery from experiencing a short circuit. On the basis of the above considerations, an alkaline zinc–iron flow battery with the membrane affords stable performance for ~240 cycles, free of zinc dendrite/accumulation, at current densities ranging from 80 to 160 mA cm$^{-2}$. Furthermore, 8 h and 7 h plating/stripping experiments at 40 mA cm$^{-2}$ yield an average coulombic efficiency (CE) of 96.54%, an energy efficiency (EE) of 91.92%, and an areal discharge capacity above 130 mAh cm$^{-2}$. A peak power density of 1056 mW cm$^{-2}$ is achieved at 1040 mA cm$^{-2}$ at 50% state-of-charge (SOC). Unlike traditional ion exchange membranes, the nanoporous membranes, isolating redox-active species from

charge-balancing ions through pore size exclusion[11–13], have high chemical stability in both strong acid (or base) and a strongly oxidizing medium, which provides a strategy toward the design and fabrication of high-performance membranes for alkaline zinc-based batteries.

## Results

**Optimization of structure and performance of the membranes.** To realize this idea, a nanoporous poly (ether sulfone)/sulfonated poly (ether ether ketone) (PES/SPEEK) membrane[14] (see Supplementary Figure 1 for chemical structure and ${}^1$H-NMR) is selected due to its easily tunable pore size and charge (Supplementary Figure 2). The charge density of the membranes was tuned by changing the content of SPEEK in the cast solution, where the SPEEK content in the polymer was 0 wt% (denoted as P0), 15 wt% (denoted as P15), 20 wt% (denoted as P20), and 25 wt% (denoted as P25). The cross-section (Fig. 2) and surface morphologies (Supplementary Figure 3) of the nanoporous P0, P15, P20, and P25 membranes were recorded by a field-emission scanning electron microscope (FE-SEM, JSM-7800F). As shown in Fig. 2, the cross-section of P0, P15, P20, and P25 all demonstrate a similar sponge-like, porous structure, with nano-size cells separated by ultrathin walls. With increasing SPEEK content in the cast solution, no distinct difference could be found on the structures of these membranes. While further magnifying the surface of these membranes, nanoscale pores can be found obviously, as evidenced in Supplementary Figure 3. This sponge-like, porous structure is thus expected to lead to a fast transportation of charge-balancing ions ($K^+$, $Na^+$, and hydroxyl ion simultaneously) through the membrane, further decreasing the ohmic resistance of the battery. Supplementary Figure 4 shows the $N_2$ sorption isotherms and pore size distributions of the prepared nanoporous membranes by Barrett–Joyner–Halenda analysis. All the prepared nanoporous membranes demonstrated a similar pore size distribution within a narrow range of 2–7 nm. The porosity of the prepared nanoporous membranes was given in Supplementary Figure 5a, where the P0 membrane exhibited the highest porosity among the prepared nanoporous membranes, while the P15, P20, and P25 presented a similar porosity. Different from the tendency of porosity, the P15, P20, and P25 membranes exhibited an increased electrolyte uptake for both positive and negative electrolytes (Supplementary Figure 5b), whereas the electrolyte uptake of P0 membrane is in the range between the electrolyte uptake of P15 and P20 membrane. The increased electrolyte uptake was mainly attributed to the increased hydrophilic SPEEK content in the membrane, which would integrate more alkaline solution in the membrane.

In a flow battery, the permeability of active species through the membrane will induce the self-discharge and capacity decay. Therefore, the permeability of active species will have great influence on the battery performance. Due to the fact that the diffusion coefficient of $Fe(CN)_6^{3-}$ is higher than that of $Fe(CN)_6^{4-}$[15], the permeability of $Fe(CN)_6^{3-}$ is thus measured to link the ion selectivity of the nanoporous membranes with their morphology. To confirm the mutual repulsion effect of the prepared nanoporous membranes on the negatively charged zincate ions, the permeability of $Zn(OH)_4^{2-}$ is measured as well. As displayed in Supplementary Figure 6a and b, all the membranes demonstrated extremely low permeability for both $Fe(CN)_6^{3-}$ (the permeability rate of $Fe(CN)_6^{3-}$ for P0, P15, P20, and P25, calculated according to Fick's diffusion law, was $5.39 \times 10^{-5}$ cm$^2$ h$^{-1}$, $0.582 \times 10^{-5}$ cm$^2$ h$^{-1}$, $3.48 \times 10^{-5}$ cm$^2$ h$^{-1}$, and $4.12 \times 10^{-5}$ cm$^2$ h$^{-1}$, respectively) and $Zn(OH)_4^{2-}$ anions, which is expected to obtain a high battery performance.

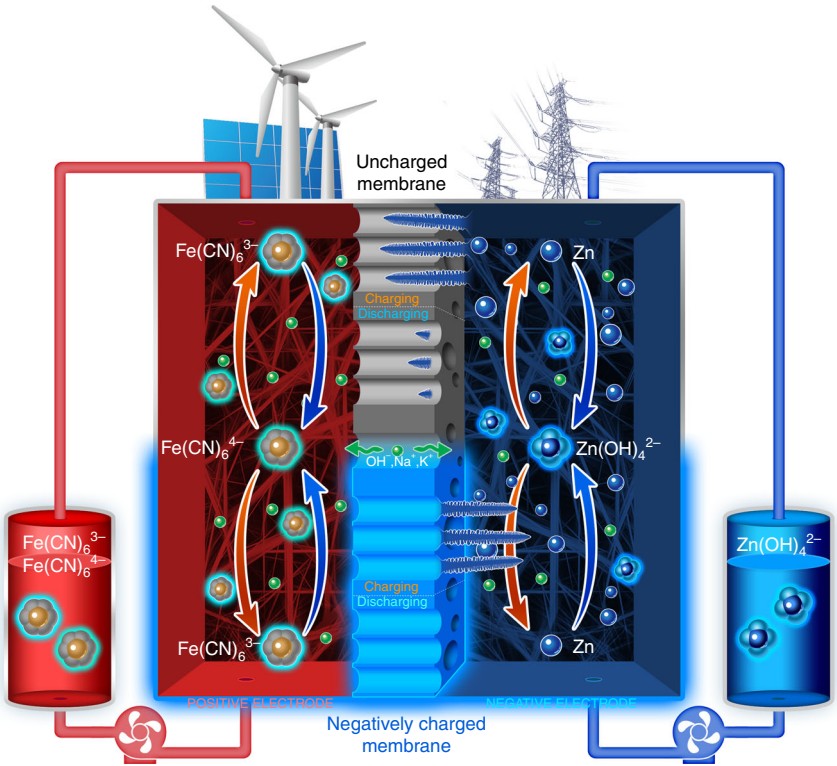

**Fig. 1** Schematic of dendrite-free alkaline zinc–iron flow battery. The schematic represents the zinc dendrite/accumulation of the zinc–iron flow battery when employing an uncharged (top) and a negatively charged (bottom) nanoporous membrane

To further demonstrate that a nanoporous membrane can be utilized in aqueous flow battery successfully, the battery performance of the prepared nanoporous membranes was investigated by using an alkaline zinc–iron flow battery. Supplementary Figure 7 shows the efficiencies of the alkaline zinc–iron flow battery with P15, P20, P25, and P0 membranes. It can be found that all the batteries show a CE of above 99% at the current density of 80 mA cm$^{-2}$, while the VE increased from 85 to 89% with increasing SPEEK content, which is well in accordance with the results of ion conductivity (Supplementary Figure 8a) and EIS measurement (Supplementary Figure 8b). The higher SPEEK content in the polymer will lead to a more continuous pore for the membrane, and simultaneously endow the membrane with a higher content of the sulfonic acid group, which is beneficial for the transportation of charge-balancing ions (Supplementary Figure 9). Consequently, a battery with a P20 membrane affords an optimized performance. In the following section, P20 membrane was selected as an example for further study. For comparison, a battery with a P0 membrane with a similar morphology delivers a CE of 99% and a VE of 87% at the same condition.

**Zinc dendrite/accumulation-free performance**. To show that the zinc dendrite can be tackled by the negatively charged nanoporous membrane, the cycling performance of an alkaline zinc–iron flow battery employing P20 was assessed. For comparison, the performance of an alkaline zinc–iron flow battery using an uncharged P0 membrane was also investigated. Figure 3a shows the voltage profiles of the alkaline zinc–iron flow batteries using P20 and P0 membranes at 80 mA cm$^{-2}$. Distinctly, the battery with a negatively charged P20 membrane demonstrated a stable cycling performance for >70 h, while the battery with an uncharged P0 membrane only operated for ~10 h and a severe battery pulverization occurred after 13 h (Fig. 3a–c).

Looking into the voltage profiles in Fig. 3c, apart from the battery polarization, the discharge duration of the battery with an uncharged P0 membrane was tapering off (Fig. 3c), thus resulting in a decreased discharge capacity of the battery (Fig. 3d). By contrast, for a battery with a negatively charged P20 membrane, a stable plating/stripping process for 150 cycles without capacity decay was achieved, evidencing a striking stability because of the negatively charged sulfonic acid groups on the pore walls. To further verify the inhibition of the negatively charged nanoporous membrane for zinc dendrite and sequentially affording an improved cycling stability, an accelerated cycling experiment was performed by increasing the SOC of the battery to 85%. The higher SOC means longer charging–discharging time as well as more serious zinc dendrite, which would definitely accelerate the damage of the membrane caused by the zinc dendrite and further leading to the battery failure. As a consequence, the capacity of the battery with a P0 membrane operated at 85% SOC decreased dramatically within 20 cycles (Supplementary Figure 10), which is less stable than does a battery with the same membrane operated at a low SOC (Fig. 3d). While for the negatively charged nanoporous membranes, the negatively charged groups on the surface and pore walls could effectively repel negatively charged zincate ions through the Donnan exclusion mechanism, which means that the more negative charges on the surface and pore walls of the nanoporous membrane, the higher cycling stability of the battery can be achieved. As a consequence, an alkaline zinc–iron flow battery with a P25 membrane demonstrated a stable performance over more than 400 cycles even at 85% SOC, which is much more stable than does a battery with a P15 membrane at the same condition (Supplementary Figure 10).

Field-emission scanning electron microscopy (FE-SEM, JSM-7800F) and energy-dispersive X-ray spectroscopy (EDS) were used to analyze the morphology and component of the negative electrode of the batteries with P20 and P0 membranes. Figure 4a showed an optical image of the negative electrode of the battery

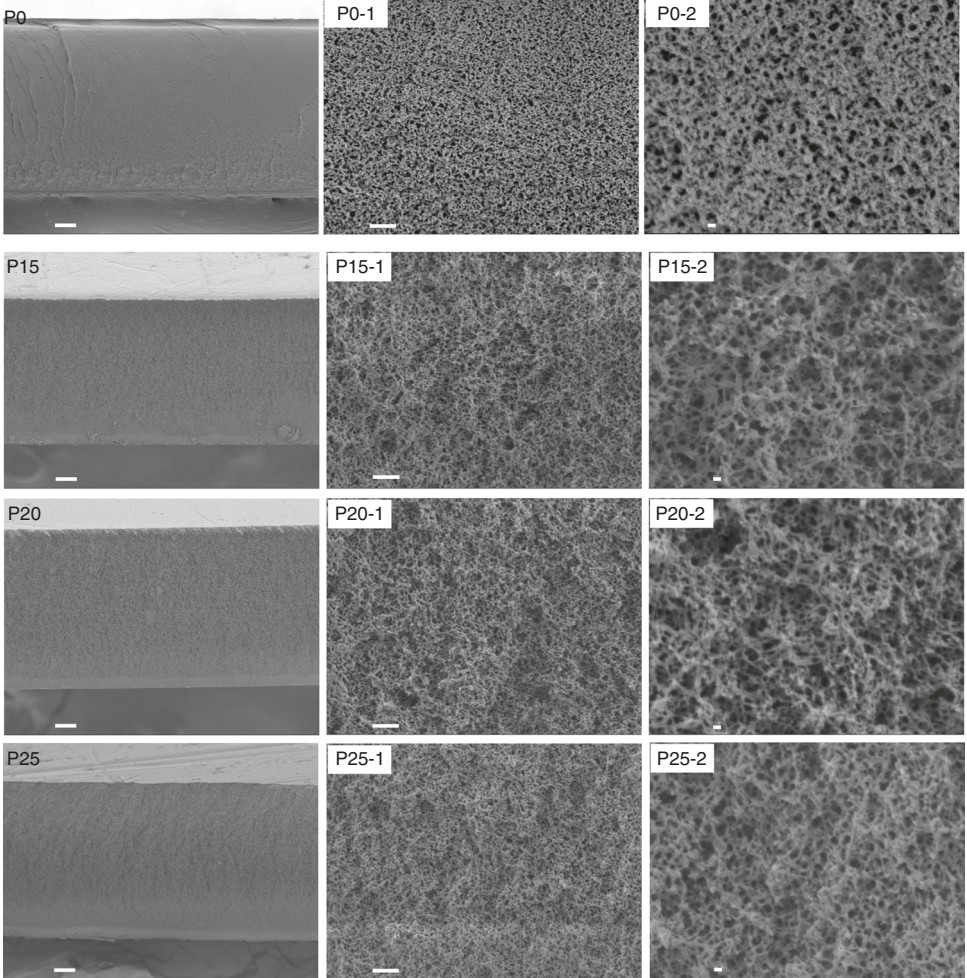

**Fig. 2** Cross-section morphologies of the prepared membranes. In PX, X represents the sulfonated poly(ether ether ketone) (SPEEK) content in the polymer (for instance, the SPEEK content in the polymer P20 is 20 wt%). PX-1 and PX-2 are the different magnifications of PX. The scale bars for PX, PX-1, and PX-2 are 10 μm, 1 μm, and 100 nm, respectively

that was assembled with a P0 membrane at the end of the 53rd cycle of charging. Formation of zinc dendrite with acerose type was demonstrated from Fig. 4b, c. During the charging process, the negatively charged zincate ions in the electrolyte were deposited irregularly, when the zinc crystal nucleus were formed, either in the membrane direction or inner carbon felt electrode direction (Fig. 1). The deposited zinc in the membrane direction can be easily piercing into the membrane (Supplementary Figure 11a–c) and even worse, shorting the battery. During the discharging process, the successive zinc metal (dendrite) was stripped gradually and became disconnected, leaving some zinc metal in the carbon felt as well as in the membrane (Fig. 1). The zinc metal (dendrite) in carbon felt can continue to be stripped, while those pierced into the membrane cannot be utilized any more during discharge (Fig. 4e–h, EDS in Fig. 4g, h confirmed that the lumps in P0 membrane were metallic zinc), resulting in a decreased discharge capacity of the battery (Fig. 3d). This in turn lowered the concentration of zincate ions (active material) in the negative electrolyte, and further resulting in a concentration polarization of the battery. Moreover, the zinc metal left in the membrane blocked the ion transport channel, thus impeding the transportation of charge-balancing ions through the membrane and further increasing the membrane resistance. The increased membrane resistance will result in zinc accumulation when the battery was discharging (Supplementary Figure 12a–c), further decreasing the discharge capacity of the battery. The

concentration polarization along with the increased membrane resistance thus resulted in a high battery polarization at the end of charge, which is consistent with the voltage profile of the battery with a P0 membrane (Fig. 3c).

By contrast, when the same procedure is utilized to a negatively charged P20 membrane, at the end of the 183rd cycle of charging, the morphology of the deposited zinc is smooth as shown in Fig. 4i–l. Different from the charging process of a battery using a P0 membrane, the negatively charged zincate ions are mostly deposited in the inner carbon felt electrode direction when the battery used a P20 membrane (Fig. 1), since the pore walls and the surface of a P20 membrane carried negatively charged sulfonic acid groups and thereby repelling zincate ions depositing in the membrane direction. A smooth membrane surface at the end of both charging (Supplementary Figure 11d–f) and discharging (Fig. 4m, n) was found, and no zinc element on the membrane surface can be detected by the EDS as well (Supplementary Figure 11f and Fig. 4o), further confirming the above speculations. Another advantage for zincate ions depositing in the inner carbon felt electrode was that the deposited zinc metal could form a well-conductive network with the carbon felt, hence affording a carbon felt/metallic zinc composite electrode. During the discharge process, the metallic zinc in the composite electrode can be fully utilized because of the well-conductive network between the metallic zinc and the carbon felt, thereby delivering a stable discharge capacity (Fig. 3a, d) and a bare

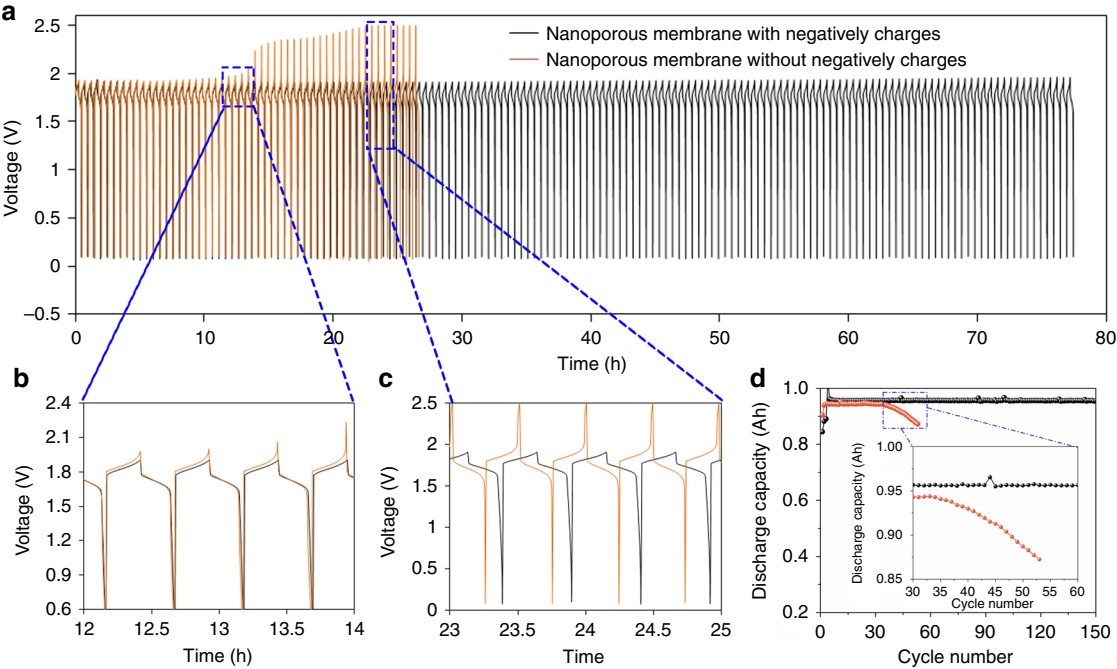

**Fig. 3** Electrochemical performance of the alkaline zinc–iron flow battery. **a** Cycling performance of the alkaline zinc–iron flow battery with a P20 and a P0 membrane at 80 mA cm$^{-2}$ with 30 min for each plating/stripping process. The charge process was controlled by the charge time (15 min) to keep a constant charge capacity. **b**, **c** Detailed voltage profiles marked by blue rectangles in panel **a**. **d** Comparison of discharge capacity of the alkaline zinc–iron flow batteries with P20 and P0 membranes. P20 is a negatively charged nanoporous membrane, where the sulfonated poly(ether ether ketone) (SPEEK) content in the polymer is 20 wt%; P0 is an uncharged nanoporous membrane, where the SPEEK content in the polymer is 0 wt%

carbon felt electrode as ever without no obvious zinc accumulation (Supplementary Figure 12d–f). Benefiting from the negatively charged sulfonic acid groups on the pore walls, this kind of membrane can thus effectively tackle the issue of zinc dendrite, even the zinc accumulation for alkaline zinc–iron flow battery.

**Quantum chemistry calculations**. Molecular dynamics (MD) simulations were employed to calculate both $Zn(OH)_4^{2-}$ rejection and ion transport through the designed membrane. The structures of the porous membranes (P0 and P20 were employed as model materials) were built using two kinds of fragmented cluster models, respectively, which were saturated by protons (Supplementary Figure 13 and Supplementary Table 1 (P20)). The final structures and accessible solvent surface models of P0 and P20 were illustrated in Supplementary Figure 14a–b and Fig. 5a, b. MD simulations (Supplementary Movie 2 (P20)) clearly show that P20 with negative charges could reject $Zn(OH)_4^{2-}$ anions very well, while allowing the permeability of Na$^+$. This is also illustrated in Fig. 5c where snapshots of the system using P20 have been taken at different times. Water molecules and OH$^-$ in the snapshots are removed for clarity. These snapshots also show that the positively charged Na$^+$ could easily penetrate through the membrane, which accounts for the higher VE of the battery with a P20 membrane than does a battery with a P0 membrane. By contrast, no significant rejection between P0 membrane and $Zn(OH)_4^{2-}$ anions could be demonstrated in both Supplementary Movie 1 (P0) and the snapshots in Fig. 4c. And the amount of Na$^+$ penetrated through the P0 membrane is much less than those of P20 membrane (Fig. 5c). Figure 5d and e exhibit the statistical distribution of Na$^+$ and $Zn(OH)_4^{2-}$ using P0 and P20 membrane, respectively. The distance between $Zn(OH)_4^{2-}$ anions and P20 membrane (Fig. 5e) is much longer than the distance between $Zn(OH)_4^{2-}$ anions and P0 membrane (Fig. 5d) because of the charge repulsion effect. The longer distance between $Zn(OH)_4^{2-}$ anions and the P20 membrane means that the

deposition of $Zn(OH)_4^{2-}$ anions in the membrane direction is much more difficult than those in the backward direction of the membrane since the diffusion of $Zn(OH)_4^{2-}$ anions into P20 membrane surface is prohibited, thus affording a very smooth zinc morphology (or zinc dendrite free, Fig. 4j–l). In comparison, from the simulations (Supplementary Movie 1 (P0), Fig. 5c, d), it can be found that $Zn(OH)_4^{2-}$ anions can be easily diffused into P0 membrane surface, which would definitely result in $Zn(OH)_4^{2-}$ anions depositing in membrane direction, along with the formation of acerose-type zinc dendrite (Fig. 4b–d).

**Assessment of cycling performance**. Prompted by the negatively charged sulfonic acid group on the pore walls, a battery with this kind of membrane is expected to deliver a good stability as well as good electrochemical performance. Figure 6a shows that a battery with a P20 membrane affords an average CE of 99.64%, an average EE of 87.72% at a current density of 80 mA cm$^{-2}$ and an average CE of 99.67%, and an average EE of 78.67% at a current density of 160 mA cm$^{-2}$, demonstrating a stable cycling performance for about 240 cycles. Working at a current density as high as 160 mA cm$^{-2}$ with a stable cycling performance has been rarely reported among the recently reported flow batteries, especially for the zinc-based flow batteries, since the higher working current density normally results in more serious zinc dendrite. More importantly, the battery exhibits an electrolyte utilization of nearly 75% at a theoretical capacity of 21.44 Ah L$^{-1}$ for each cycle (Fig. 6b) even at a high current density of 160 mA cm$^{-2}$.

To further validate the advantages of the negatively charged nanoporous membranes on tackling zinc dendrite and accumulation for alkaline zinc–iron flow battery, 8 h and 7 h plating/stripping experiments were performed at 40 mA cm$^{-2}$. An alkaline zinc–iron flow battery with a P20 can afford a stable cycling performance at 40 mA cm$^{-2}$ for nearly 8 h or 7 h for each plating/stripping step (Fig. 6c). An average CE of

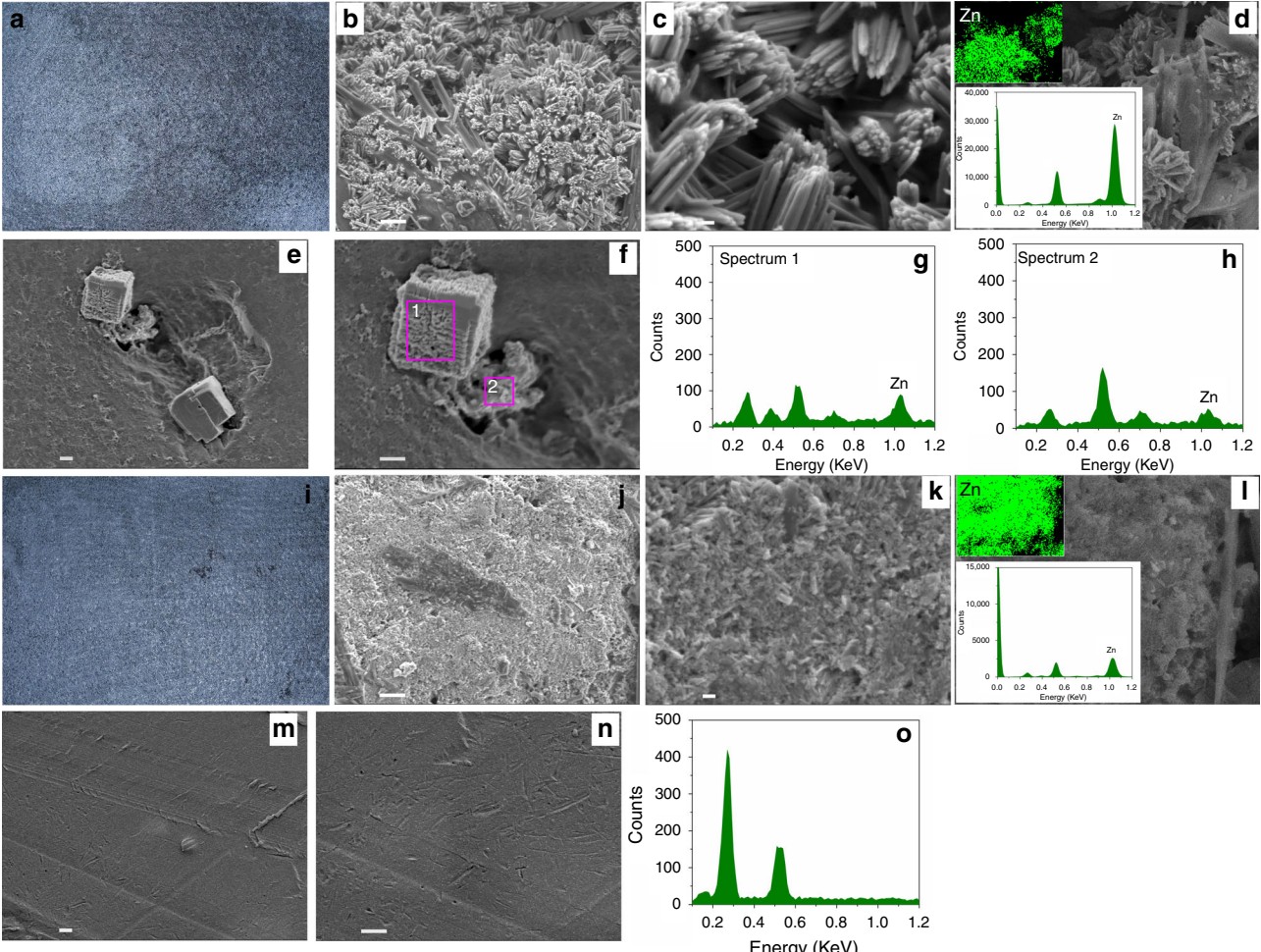

**Fig. 4** Morphologies of zinc metal (dendrite) and membrane surface. **a** Optical image of the negative electrode at the end of 53rd charge for the alkaline zinc–iron flow battery assembled with a P0 membrane. **b** SEM image of zinc metal (dendrite) in the carbon felt in panel **a**. **c** Magnified SEM image of zinc metal (dendrite) in panel **b**. **d** EDS spectrum and EDS mapping of the zinc metal (dendrite) in the corresponding SEM image. **e** The surface morphology of P0 membrane at the end of 65th discharge. **f** Magnified surface morphology of P0 in panel **e**. **g**, **h** Corresponding EDS spectrums of the lumps in P0 membrane marked by mauve rectangle in panel **f**, where zinc metal (dendrite) was pierced into P0 membrane and retained in the membrane even at the end of discharge. **i** Optical image of the negative electrode at the end of 183rd charge for the alkaline zinc–iron flow battery assembled with a P20 membrane. **j** SEM image of zinc metal in the carbon felt in panel **i**. **k** Magnified SEM image of zinc metal in panel **j**. **l** EDS spectrum and EDS mapping of the zinc metal in the corresponding SEM image. **m** The surface morphology of P20 membrane at the end of 186th discharge. **n** Magnified surface morphology of P20 in panel **m**. **o** Corresponding EDS spectra of P20 membrane in panel **n**, where no zinc metal (dendrite) can be found at the end of discharge. The battery is charged and discharged at 80 mA cm$^{-2}$. P20 is a negatively charged nanoporous membrane, where the sulfonated poly(ether ether ketone) (SPEEK) content in the polymer is 20 wt%; P0 is an uncharged nanoporous membrane, where the SPEEK content in the polymer is 0 wt%. SEM scanning electron microscopy, EDS energy-dispersive X-ray spectroscopy. Scale bars of **b**, **e**, **f**, **j**, **m**, and **n** are 1 μm; scale bars of **c** and **k** are 100 nm

96.54% and EE of 91.92% was achieved over 27 cycles (withstanding nearly a 200-h cycling test) with flat plating/stripping curves, which is unprecedented for the zinc-based flow batteries to the best of our knowledge. The long plating/stripping process thus in turn leads to a stable areal discharge capacity of 154 mAh cm$^{-2}$ for 8 h and 135 mAh cm$^{-2}$ for 7 h, with a discharge energy of 272 mWh cm$^{-2}$ for 8 h, and 238 mWh cm$^{-2}$ for 7 h (Fig. 6d). To the best of our knowledge, this areal capacity is the highest among recently reported zinc-based flow batteries. Even under a prolonged plating process (or high areal capacity), the morphology of the deposited zinc metal on the carbon felt remains smooth and no distinct zinc dendrite could be found (Supplementary Figure 15a–c), supporting that a negatively charged porous membrane can tackle zinc dendrite efficiently for alkaline zinc–iron flow battery. As a consequence, the membrane facing the negative side still exhibited a very flat and smooth surface as ever (Supplementary Figure 15d). Moreover, no obvious zinc accumulation could be examined on the carbon felt (Supplementary Figure 15e–g) and the membrane surface toward the negative side was kept smooth (Supplementary Figure 15h) as well after stripping, demonstrating that the negatively charged zincate ions were forced to deposit in the inner carbon felt direction only by the negative charges on the pore walls and forming a well-conductive network between the deposited zinc metal and the carbon felt. Judging from the above results, the porous membrane endowed with negative charge on the pore walls can direct a pathway for addressing the issue of zinc dendrite and zinc accumulation for alkalize zinc–iron flow battery and simultaneously affording the battery with excellent performance in a wide current density range in terms of both selectivity and conductivity.

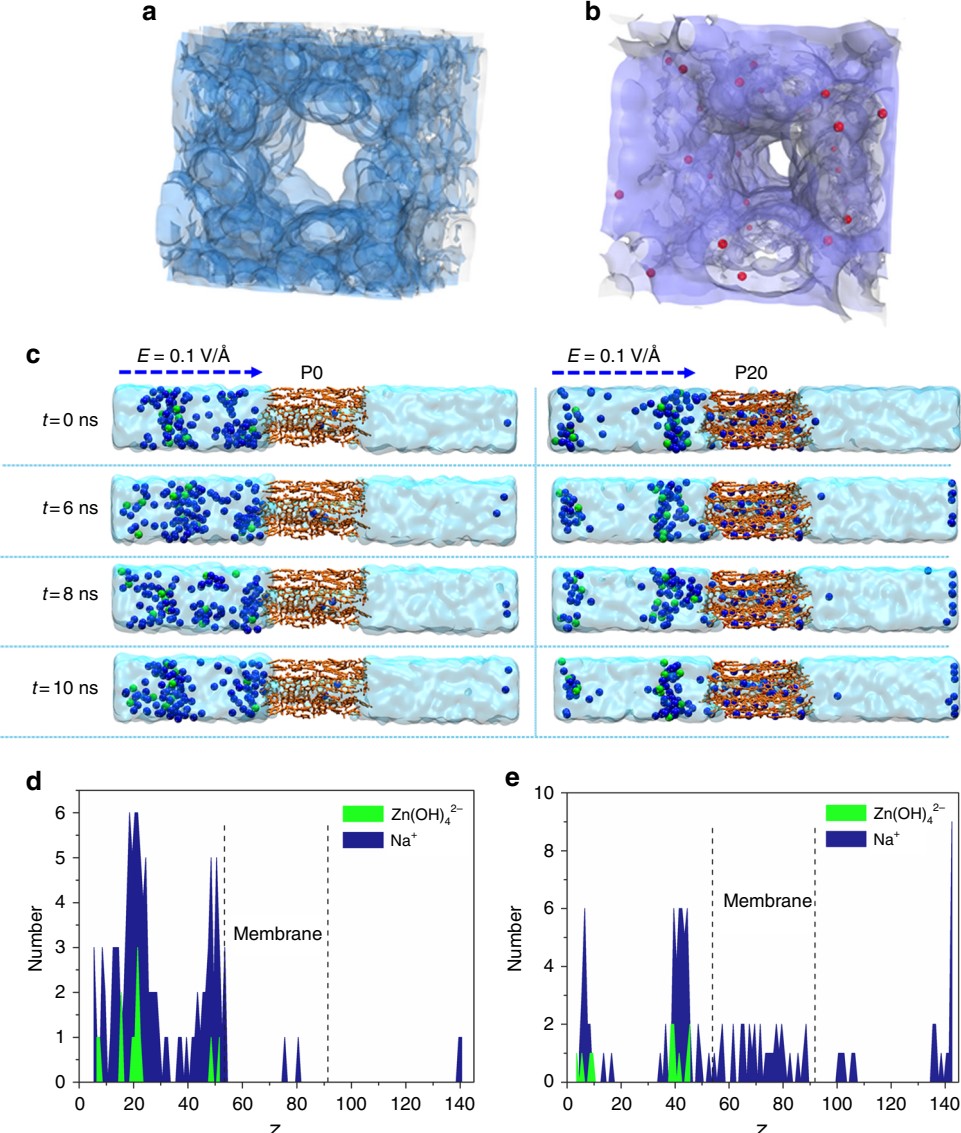

**Fig. 5** Molecular dynamics simulation results using different membranes. **a** Accessible solvent surface model of P0. **b** Accessible solvent surface model of P20. The red spheres represent sulfonic acid groups. **c** Snapshots of the P0 and P20 membranes at different times, taken at 0, 6, 8, and 10 ns in an electric field (E) of 0.1 V Å$^{-1}$. The blue spheres represent Na$^+$ and the green spheres represent Zn(OH)$_4^{2-}$. **d** Statistical distribution of Na$^+$ and Zn(OH)$_4^{2-}$ at 10 ns using P0 membrane. **e** Statistical distribution of Na$^+$ and Zn(OH)$_4^{2-}$ at 10 ns using P20 membrane. P20 is a negatively charged nanoporous membrane, where the sulfonated poly(ether ether ketone) (SPEEK) content in the polymer is 20 wt%; P0 is an uncharged nanoporous membrane, where the SPEEK content in the polymer is 0 wt%

**Power density of the alkaline zinc–iron flow battery**. The rate performance is one of the most important factors to evaluate the merit of an alkaline zinc-based flow battery. Figure 7a shows the performance of the alkaline zinc–iron flow battery with a P20 membrane operating at the current density ranging from 80 to 160 mA cm$^{-2}$. A slightly increased CE can be found with increasing current density, while the VE slightly decreased owing to the increased electrochemical polarization and ohmic polarization. Overall, the alkaline zinc–iron flow battery with a P20 membrane demonstrates an excellent rate performance. Beyond that, a battery with a P20 membrane delivers an open-circuit voltage (OCV) of 1.81 V at 50% SOC. With SOC increasing from 5 to 95%, the OCV increased from 1.72 to 1.93 V monotonically as displayed in Fig. 7b. This high OCV in combination with a high working current density is thus expected to afford the battery with an excellent power density.

Figure 7c shows the polarization curves of an alkaline zinc–iron flow battery with a P20 membrane at 20%, 50%, and 80% SOC, respectively. With beneficial fast redox kinetics of the Zn(OH)$_4^{2-}$/Zn and Fe(CN)$_6^{3-}$/Fe(CN)$_6^{4-}$ couples in alkaline medium, the battery displayed no signs of activation polarization. And the concentration polarization was not observed as well because of the high electrolyte flow rate. With the above advantages, a peak power density of 787 mW cm$^{-2}$ was achieved at a current density of 760 mA cm$^{-2}$ even at low SOC (20% SOC). This value is much higher than that of a reported zinc–iron flow battery (676 mW cm$^{-2}$ at a current density of 660 mA cm$^{-2}$ at 70% SOC)[6]. A higher peak power density (1056 mW cm$^{-2}$) was delivered at a current density of 1040 mA cm$^{-2}$ at a higher SOC (50%), which is among the highest for recently reported flow battery systems[6,8,15–21] (Table 1) and even higher than some traditional flow battery systems. It should be noted that the peak power density of the battery at 80% SOC cannot be obtained since

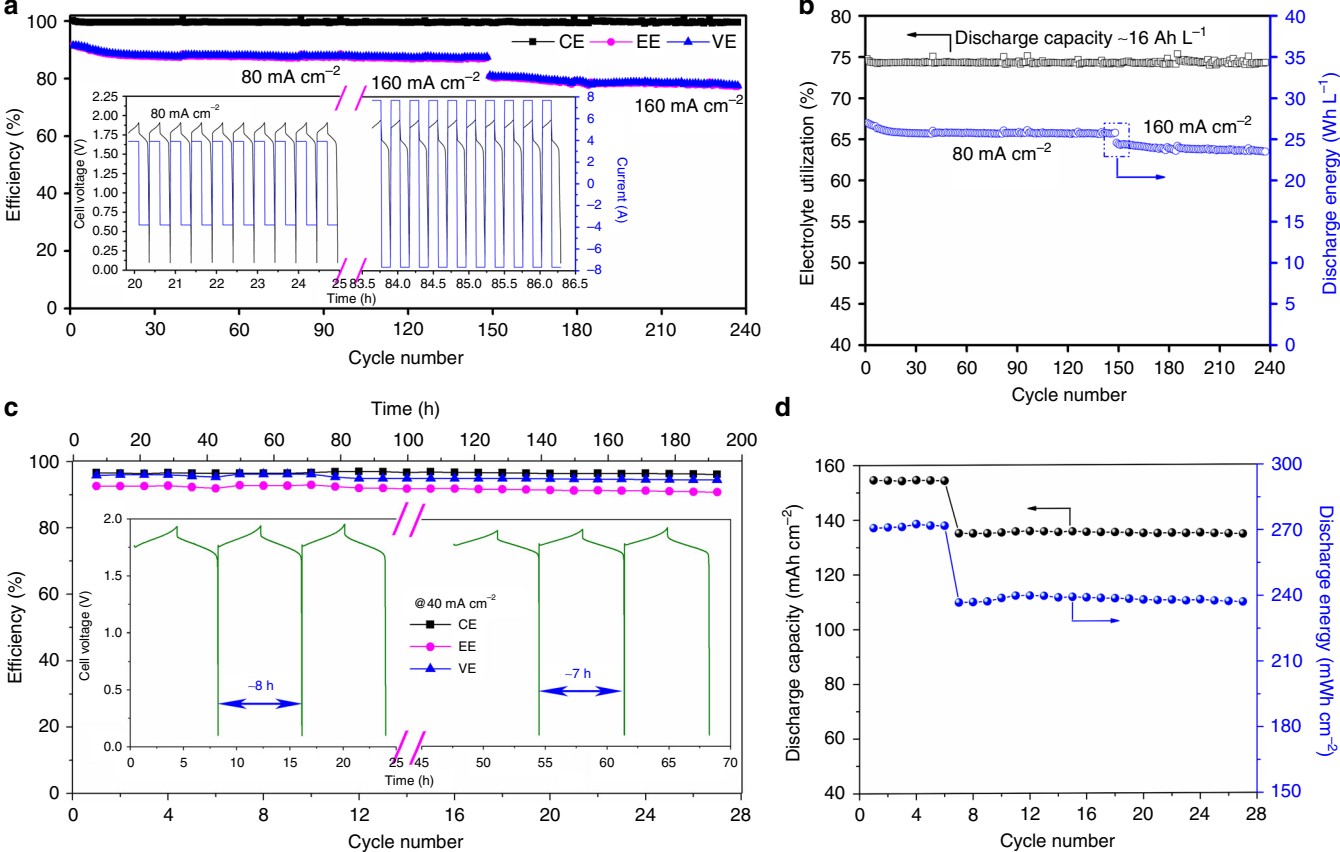

**Fig. 6** Cycling performance of the alkaline zinc–iron flow battery. **a** Cycle performance of the alkaline zinc–iron flow battery using a P20 membrane. Insets: representative charge and discharge profiles. **b** Electrolyte utilization and discharge energy during the cycling. **c** Cycle performance of the alkaline zinc–iron flow battery at a current density of 40 mA cm$^{-2}$. Inset: representative charge and discharge curves of the alkaline zinc–iron flow battery. The charge time was kept for 4 h between the first and sixth cycles and 3.5 h between the 7th and 27th cycles. **d** Corresponding discharge capacity and discharge energy for each cycle. CE coulombic efficiency, EE energy efficiency, VE voltage efficiency; P20 is a negatively charged nanoporous membrane, where the sulfonated poly(ether ether ketone) (SPEEK) content in the polymer is 20 wt%

the current has run out of the full scale of the device (ArbinBT 2000, 5 V, 10 A). Moreover, even at a high current density of 840 mA cm$^{-2}$, the battery can still afford a voltage of 1.3 V, which is comparable to the reversible voltage of many advanced flow battery systems. Taken together, in combination with a nanoporous P20 membrane with an alkaline zinc–iron flow battery, the present study directs a pathway for addressing the issue of both ohmic resistance and cost brought from Nafion series ion exchange membranes for the newly developed aqueous flow battery systems and accelerating a step for these batteries moving forward.

## Discussion

In summary, we have designed and fabricated a nanoporous membrane with negative charge for tackling the zinc dendrite/ accumulation of an alkaline zinc-based flow battery. By employing the negatively charged porous membrane, an alkaline zinc–iron flow battery demonstrated a stable performance free of zinc dendrites in a wide range of working current densities. Due to the mutual repulsion between the negatively charged zincate ions in the alkaline medium and the negatively charged pore walls in the nanoporous membrane, the plating process was forced to take place mostly in the inner carbon felt direction, forming a smooth metallic zinc and leaving the membrane intact. Given this unique advantage, an alkaline zinc–iron flow battery can deliver a high areal discharge capacity. Furthermore, a battery with the prepared nanoporous membrane affords a very high peak power

density due to the fast transport of ions through the membrane. This work directs a pathway for further research on other alkaline zinc-based batteries.

## Methods

**Materials**. Poly(ether ether ketone) (PEEK) and poly(ether sulfone) (PES) were offered by Changchun Jilin University Special Plastic Engineering Research. Sulfonated poly(ether ether ketone) (SPEEK) with a degree of sulfonation (DS) of around 0.78, calculated by $^1$H-NMR, was prepared by direct sulfonation of PEEK with sulfuric acid at 70 °C for 2 h, as described elsewhere[22]. DS can be calculated from the proton integration, as given in equation (1)

$$\frac{DS}{12 - 2DS} = \frac{A_{H_1}}{\sum A_{H_N}}$$
(1)

where $A_{H_1}$ represents the integration area of H$_1$ peak, $\sum A_{H_N}$ indicates the integration area of all proton peaks.

Polyvinylpyrrolidone (PVP) with a molecular weight of 58,000 was used as received. Potassium hydroxide, N,N-dimethylacetamide (DMAc) were purchased from Tianjin Damao Chemical Reagent Factory. Sodium hydroxide was purchased from Tianli Chemical Reagent Co., Ltd. Sodium ferrocyanide was bought from Sinopharm Chemical Regent Co., Ltd. Zinc oxide was bought from Kermel Chemical Reagent Factory. These reagents were supplied with analytical grade. Other reagents were bought from Sigma Aldrich unless stated otherwise and used as received.

**Preparation of nanoporous membranes**. The nanoporous membranes with and without negative charge were prepared by the phase inversion technique. To design membranes with negative charge, PES polymers were selected as the matrix, SPEEK was selected to tune the membrane morphology and offer the membrane with negative charges. The polymers were dissolved in DMAc to form a 35 wt% solution, the SPEEK content in the polymer were 15 wt%, 20 wt%, and 25 wt%, respectively.

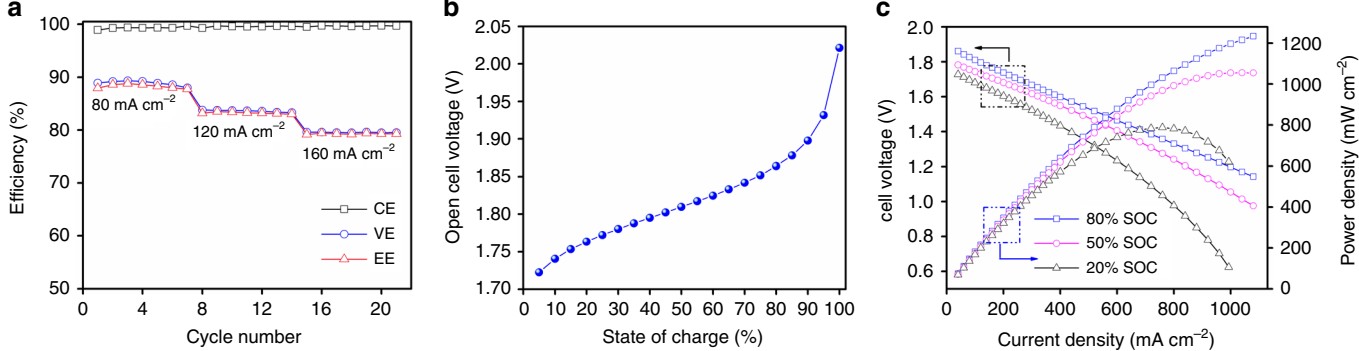

**Fig. 7** Battery performance of the alkaline zinc–iron flow battery. **a** The cycling performance of the alkaline zinc–iron flow battery employing a P20 membrane at different current densities. **b** The open-cell voltage versus state-of-charge of an alkaline zinc–iron flow battery using a P20 membrane. The above test was conducted in a battery with an active area of 48 cm². **c** The polarization of the alkaline zinc–iron flow battery at 80% SOC, 50% SOC, and 20% SOC using a P20 membrane. The polarization of the alkaline zinc–iron flow battery was investigated using a battery with active area of 9 cm². SOC state-of-charge; P20 is a negatively charged nanoporous membrane, where the sulfonated poly(ether ether ketone) (SPEEK) content in the polymer is 20 wt%; CE coulombic efficiency, EE energy efficiency, VE voltage efficiency

**Table 1 Power density for different flow battery systems**

| Flow battery systems | State of charge (%) | Current density (mA cm⁻²) | Peak power density (mW cm⁻²) | Reference |
|---|---|---|---|---|
| Zinc–iodine–bromide flow battery | Not given | 70 | 50 | Ref. 8 |
| Organic–organometallic flow battery | 90 | 150 | 60 | Ref. 16 |
| (Ferrocenylmethyl) trimethylammonium chloride methyl viologen flow battery | 100 | 200 | 125 | Ref. 18 |
| Flavin mononucleotide-based flow battery | Not given | 300 | 160 | Ref. 19 |
| All-soluble all-iron flow battery | 70 | 200 | 160 | Ref. 15 |
| Alloxazine-based flow battery | 100 | 580 | 350 | Ref. 20 |
| Alkaline quinone flow battery | 100 | 700 | 450 | Ref. 21 |
| Zinc–iron flow battery | 70 | 660 | 676 | Ref. 6 |
| Anthraquinone derivatives–bromide flow battery | 90 | 1500 | 700 | Ref. 17 |
| Alkaline zinc–iron flow battery | 50 | 1040 | 1056 | This work |

Then the solution was cast onto a clean glass plate at room temperature with humidity < 50% to avoid the penetration of water vapor into the polymer solution. Afterward, the plate was immersed into water to form the ordered nanoporous PES/SPEEK membranes. The prepared nanoporous PES/SPEEK membranes were then soaked in isopropanol for 30 min. The membranes were evaporated at room temperature for 2 h. Finally, the membranes were stored in water for use. The thickness of the prepared nanoporous PES/SPEEK membranes was 65 ± 3 μm. The prepared nanoporous PES/SPEEK membranes with different SPEEK content in the cast solution were referred to as P15, P20, and P25, respectively. A nanoporous uncharged PES membrane was employed for reference, which was prepared from a PES/PVP cast solution (the polymer concentration was 35 wt% and the PVP content in the polymer was kept as 50 wt%) to obtain a similar pore structure with a nanoporous PES/SPEEK membrane. During the procedure of phase inversion, PVP was dissolved in water, resulting in a nanoporous uncharged PES membrane. The PES membrane without charge was referred to as P0.

**Membrane and zinc metal morphologies**. The morphologies of the membrane surface and cross-section were recorded by FE-SEM (JSM-7800F), equipped with an energy-dispersive X-ray spectroscope (EDS). The EDS and EDS mapping were employed to substantiate the zinc metal (dendrite) on the carbon felt and the porous membrane after charging and discharging experiments. The cross-sections of the membranes were obtained by breaking the membranes in liquid nitrogen and coating them with gold prior to imaging. To confirm the existence of negative charges in a nanoporous P20 membrane, high-resolution transmission electron microscopy (HRTEM, FEI Tecnai G² F30S-Twin microscope, 300 kV) was performed on the membrane, which was dyed with 0.5 M AgNO₃. For comparison, a nanoporous P0 membrane, which was dyed with 0.02 M palladium chloride solution, was investigated as well. Note that the P0 membrane was dyed with 0.02 M palladium chloride solution to confirm the nonexistence of PVP in the membrane (or confirm the nonexistence of charge in the membrane). All the samples were first fixed in epoxy before being cut into thin slice samples.

**Electrolyte uptake**. Electrolyte uptake was measured by immersing the prepared porous membranes with certain quality into the electrolytes (0.4 mol L⁻¹ Zn (OH)$_4^{2-}$ + 3 mol L⁻¹ sodium hydroxide solution or 0.8 mol L⁻¹ Na₄Fe(CN)₆ + 3 mol L⁻¹ potassium hydroxide solution) at room temperature for 48 h. The membrane was then taken out, wiped with a tissue paper, and quickly weighed. The electrolyte uptake was calculated in equation (2):

$$\text{Electrolyte uptake} = \frac{(m_w - m_d)}{m_d} \times 100\% \qquad (2)$$

where $m_w$ and $m_d$ are the weights of hydrated and dry membranes, respectively.

**Membrane porosity**. The prepared porous membranes were soaked in isopropanol for 24 h first to saturate them with isopropanol. Then the weight of the saturated membrane was obtained after quickly wiping out the surface isopropanol by tissue. Finally, the samples were dried in a vacuum oven at 60 °C for 24 h and weighed. The membrane porosity is calculated in equation (3):

$$\varepsilon = \frac{(m_w - m_d)/\rho}{s \times l} \times 100\% \qquad (3)$$

where $\varepsilon$ is the porosity of the membrane, $m_w$ and $m_d$ are the mass of the wet membrane and dry membrane, respectively, $\rho$ is isopropanol density, $s$ is membrane area, and $l$ is membrane thickness.

**Area specific resistance and ionic conductivity**. The area specific resistance of a membrane was measured via a conductivity cell. The cell was filled with 0.5 M NaOH in each compartment separated by a membrane with an effective area of 1 cm². The electric resistance was measured by using an electrochemical impedance spectroscopy (EIS) over a frequency range from 1000 Hz to 1000 kHz. The area specific resistance was calculated in equation (4):

$$R = (R_1 - R_2) \times S \qquad (4)$$

where $R_1$ and $R_2$ are the electric resistances of the cell with and without a membrane, respectively, and $S$ was the effective area of the membrane.

A conductivity cell was utilized to measure the ion conductivity of the prepared porous membrane. The membranes were first immersed into 0.5 M NaCl for 24 h. The cell was assembled with a membrane and filled with 0.5 M NaCl. The effective area of the membrane is 1 cm × 1 cm. The electric resistance was measured by using EIS. The ion conductivity of the membranes was calculated as shown in equation (5):

$$\sigma = \frac{l}{s \times R} \times 100\% \tag{5}$$

where $\sigma$ is the ion conductivity of the membrane (S cm$^{-1}$), $l$ is the thickness of the membranes (cm), $s$ is the effective area of the membrane (cm$^2$), and $R$ is the membrane resistance ($\Omega$).

**Pore size and pore size distribution**. The pore size and pore size distribution of the prepared membranes were analyzed by the Brunauer–Emmett–Teller surface area analyzer (ASAP 2020, Micromeritics).

**The permeability of ferricyanide and zincate ions**. The permeability of ferricyanide ion (Fe(CN)$_6{}^{3-}$) through the prepared nanoporous membrane was determined by a diffusion cell separated by a membrane. The left cell was filled with 0.4 mol L$^{-1}$ K$_3$Fe(CN)$_6$ (or Na$_2$Zn(OH)$_4$) in 3 mol L$^{-1}$ sodium hydroxide solution (volume: 80 mL), while the right one was filled with 0.4 mol L$^{-1}$ K$_2$SO$_4$ in a 3 mol L$^{-1}$ sodium hydroxide solution (volume: 80 mL) to equalize the ionic strengths and minimize the osmotic pressure effects. Solutions in both half-cells were vigorously stirred to avoid concentration polarization. The effective area of the membrane was 9 cm$^2$. Samples of a 3-mL solution from the right cell were collected at a regular time interval. Another 3-mL fresh K$_2$SO$_4$ solution was then added to the right cell to keep the solution volume stable. The K$_3$Fe(CN)$_6$ concentration of the samples was detected using a UV-vis spectrometer. The Na$_2$Zn (OH)$_4$ concentration of the samples was detected using an inductively coupled plasma mass spectrometry (ICP-MS). The K$_3$Fe(CN)$_6$ (or Na$_2$Zn(OH)$_4$) permeability was calculated according to Fick's diffusion law as displayed in equation (6):

$$V_B \frac{dC_B(t)}{dt} = A \frac{p}{L}(C_A - C_B(t)) \tag{6}$$

where $V_B$ is the solution volume in the right reservoir, $C_B(t)$ is anionic active species concentration in the right cell as a function of time $t$, while $A$ and $L$ are the effective area and thickness of the membrane, respectively. $p$ is the permeability of anionic active species, and $C_A$ is the anionic active species concentration in the left cell.

**Hydroxyl ion permeability**. In flow batteries, the permeability of charge-balancing ions across the membrane has a great influence on ohmic resistance of a battery. The device used for measuring the permeability of hydroxyl ion is similar to a previous report[11], except that the right cell was filled with deionized water and the concentration of hydroxyl ion was characterized by Mettler Toledo pH meter.

**Battery performance**. An alkaline zinc–iron flow battery was assembled by sandwiching a membrane between two carbon felt electrodes, clamped by two graphite plates. The active area of the electrode is 6 × 8 cm$^2$. All of these components were fixed between two stainless-steel plates. Solutions consisting of 60 mL of 0.4 mol L$^{-1}$ Zn(OH)$_4{}^{2-}$ + 3 mol L$^{-1}$ sodium hydroxide solution and 60 mL of 0.8 mol L$^{-1}$ Na$_4$Fe(CN)$_6$ + 3 mol L$^{-1}$ potassium hydroxide solution were used as negative and positive electrolytes, respectively. The electrolyte was cyclically pumped through the corresponding electrodes in airtight pipelines. Charge–discharge cycling tests were conducted by ArbinBT 2000 at a constant current density ranging from 80 to 160 mA cm$^{-2}$. The charge process was controlled by the charge time to keep a constant charge capacity, while the discharge process ended with a cutoff voltage of 0.1 V. The SOC vs. OCV data were measured by incremented charging time (150 s) at 40 mA cm$^{-2}$.

Separation of power ratings and energy storage capacity is one of the attractive features for flow batteries, especially for those liquid–liquid types of flow batteries, e.g., vanadium flow battery[13], iron vanadium flow battery[23], and alkaline quinone ferrocyanide flow battery[21] (both the oxidation and reduction state of redox couples in positive and negative electrolytes are soluble in the electrolyte). The power output of these batteries was determined by the size and quantity of the battery stack, while the energy storage capacity was made up by the redox couples concentration and storage tank volume[24]. In contrast, this case was unavailable for hybrid flow batteries (e.g., zinc–bromine flow battery, zinc nickel flow battery, and zinc–iron flow battery) with a given power output, since the areal capacity of the battery was limited even if a large amount of electrolytes or a high redox couples concentration was supplied. As a consequence, the plating time of the zinc-based flow batteries was normally short, further leading to a low areal capacity. In addition, higher plating time or higher areal capacity would lead to zinc dendrites and accumulation more easily, and even worse, battery failure. Therefore, to further validate the advantageous effect of the negatively charged nanoporous membranes

on tackling zinc dendrites and accumulation for alkaline zinc–iron flow battery, an 8 h and a 7 h plating/stripping experiments were performed at 40 mA cm$^{-2}$. The active area of the electrode is 9 cm$^2$. Solutions consisting of 80 mL of 0.4 mol L$^{-1}$ Zn(OH)$_4{}^{2-}$ + 3 mol L$^{-1}$ sodium hydroxide solution and 80 mL of 0.8 mol L$^{-1}$ Na$_4$Fe(CN)$_6$ + 3 mol L$^{-1}$ potassium hydroxide solution were used as negative and positive electrolytes, respectively.

**Polarization test**. The polarization of the alkaline zinc–iron flow battery at different SOC values (20, 50, and 80%) using an ordered nanoporous PES/SPEEK membrane was tested with the same initial electrolyte compositions. Alternating discharge current density was applied and cell voltage at each current density was recorded.

**Modeling approaches**. In order to obtain the uniform pore diameter of the porous membrane, a fixed single-walled carbon nanotube (SWCNT) (length of $L = 36.89$ Å, $n = m = 3$) was inserted into a box of 20 Å × 20 Å × 38 Å filled in the open space with 17 cluster models. After building of the initial structure, the geometry of the porous membrane was refined by a self-consistent iterative procedure, and the geometry optimization was performed by means of the molecular mechanics (MM) method with the Forcite module of Materials Studio[25]. In all simulations, an orthogonal system cell with extents $(L_x, L_y, L_z) = (20, 20, \text{and } 145)$ Å was used with periodic boundary condition applying in all directions, and the studied model system was shown in Supplementary Figure 14c. First, the membrane was hydrated by filling in the open space with water, and then the hydrated membrane was placed between a reservoir of an aqueous solution with 12 Zn(OH)$_4{}^{2-}$, 100 Na$^+$, 620 water, and OH$^-$ in moderation on the left, and a reservoir of pure water with 660 water molecules on the right. A constrained graphene sheet was added to the free space between the solution reservoir and pure water reservoir, and the whole system is placed in an electric field of $|E| = 0.1$ V Å$^{-1}$ applied along the $z$ axis.

All the MD simulations were performed using the MD code large-scale atomic/molecular massively parallel simulator (LAMMPS)[26]. The all-atom optimized potential for liquid simulations (OPLS-AA) force field was used, which can capture essential many-body terms in interatomic interactions, including bond stretching, bond angle bending, van der Waals, and electrostatic interactions in this system[27]. From the perspective of atomic partial charge as input parameters, the CHELPG charge was calculated by the DFT method using Gaussian09 package at the B3LYP/Def2TZVP level (shown in Table S1)[28]. Water was described by a TIP3P model[29], where the SHAKE algorithm was used to keep the rigidity of water molecules. The van der Waals (vdW) coupling was calculated with a cutoff of 12 Å. The particle–particle–particle–mesh (PPPM) method was used to treat long-range electrostatic interactions. A 10-ns equilibrium molecular dynamics (EMD) simulation was conducted to reach the system equilibrium, and then another 10-ns non-equilibrium molecular dynamics (NEMD) simulation at the situation of the electric field was to simulate the procedure of flow battery, where the last 4 ns were used for data analysis. The NVT ensemble at $T = 298$ K for 10 ns with a time step of 1 fs was employed for all simulations.

## Data availability
The data that support the findings of this study are available from the corresponding author upon request.

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

## Acknowledgements

The authors greatly acknowledge the financial support from China Natural Science Foundation (Grant No. 21206158) and Key Project of Frontier Science, CAS (QYZDB-SSW-JSC032).

## Author contributions

Z.Z.Y., X.Q.L. and W.B.X. contributed equally to this work. Z.Z.Y. and X.Q.L. performed the experiments, analyzed the data, and wrote the initial manuscript draft. W.B.X. performed molecular dynamics simulations. Prof. X.F.L. designed and oversaw the experiments, revised, and finalized the manuscript for submission. Prof. H.M.Z. and Y.Q.D. participated in the discussions of the results. All authors reviewed the manuscript.

## Additional information

**Competing interests:** The authors declare that they have no competing interests.

