## [Peer Review File · Nature Communications]

Reviewers' comments:

Reviewer #1 (Remarks to the Author):

Dendrites is a long standing issue for batteries involving stripping/plating process, like lithium based batteries and zinc based batteries, which prevents the practical application of these batteries. This manuscript reported a dendrites free alkaline zinc-iron flow battery with long cycle life by employing a negatively charged nanoporous membrane. The innovative structure design in membrane, which performed significantly better chemical stability than traditional anion exchange membranes in alkaline medium, has produced an excellent battery performance even at a high current density of 160 mA cm⁻². This is a ground breaking work in the field of alkaline zinc based batteries from a well-respected group. I am sure the effort and novelty of this work are to be commended, which will have immediate and long-lasting impact for the zinc based batteries.

Therefore, I recommend the manuscript to be accepted and published in Nature Communications. Some minor comments are as following:

1. In part of materials, the DS of SPEEK was calculated by ¹H-NMR, the calculation method including the NMR spectrum should be supplied.
2. The permeability rate of Fe(CN)₆³⁻ should be provided.
3. From Figure 1, the SEM image of all the membranes demonstrate a sponge-like, symmetric structure, I just wondering why a porous membrane with asymmetric structure was not used.
4. For STEM measurement, why P20 and P0 were treated in a different way?
5. For better understanding the excellent performance of the battery with a negatively charged nanoporous membrane, I suggest moving Table S2 into the main article.
6. In Figure 5c, why the peak power density of the battery at 80% SOC was not given further?

Reviewer #2 (Remarks to the Author):

This work reports a dendrite-free Zn deposition induced by a negatively charged nanoporous membrane. A high cycling stability of alkaline Zn/Fe battery is demonstrated. The authors claim that the rejection of zincate anion from the membrane due to the coulombic interaction.

The major concern on this work is that it does not provide a theoretical basis for this behavior. Why does the zincate rejection prevent dendrite formation? The current explanation is not enough. In addition, a direct evidence for zincate rejection is needed.

The other concern is that many factors affecting Zn deposition are neglected or uncontrolled. Pore size and distribution are not intensively characterized for the membranes. Also water uptake and wettability can be different. It should be clarified that the ions cannot permeate through the dense matrix for the membrane with high SPEEK content (In fuel cell community, this class dense membrane is used as a proton conducting membrane). These parameters can collectively influence Zn deposition behavior. Therefore, fine controls of the structures and properties are required to prove the surface charge-induced modulation of Zn deposition. It is widely accepted in Li metal electrode technology sector that more uniform and faster Li⁺ flux through a porous membrane in contact with Li metal lead to a more uniform Li deposition. In this regard, not only the surface charge but also any difference in pore size, pore size distribution and electrolyte uptake can lead to a difference in Zn deposition behavior.

The polarization and Zn deposition are directly related with area specific resistance and transference number, respectively, which can be experimentally quantified using impedance and junction potential

measurement. Zincate diffusion, would be more important than other ion diffusions, was not investigated in this work.

The term 'backward Zn dendrite growth' is not fully verified. The observed behavior can be simply the progressive Zn deposition from the surface to the interior. If the authors stick to use the term, they may provide a theoretical basis for the backward dendrite growth.

Electrolyte uptake, porosity, ionic conductivity, area specific resistance of the series of membrane should be measured for a clearer conclusion.

Reviewer #3 (Remarks to the Author):

Comments on the paper entitled: "Negatively Charged Nanoporous Membranes: Toward Dendrites Free Alkaline Zinc Based Flow Battery with Long Cycle Life" by Zhizhang Yuan, Xiaoqi Liu, Yinqi Duan, Huamin Zhang and Xianfeng Li

This work reports on employing a negatively charged nanoporous membrane to eliminate zinc dendrites/accumulations in alkaline zinc based flow batteries. This issue is claimed to be well tackled by the repulsion effect of the negatively charged nanoporous membrane, thus resulting in a much improved performance of Zn-Fe flow batteries, in terms of cycling stability and power density. The results are interesting. However, the novelty is not high enough to meet the requirement of nature communications. First, porous membranes with and without charge on the surface of pore walls have been extensively studied for flow batteries (e.g., *Journal of Power Sources* 342, 2017, 327-334; *Journal of Power Sources* 353, 2017, 11-18; *Journal of Power Sources* 298, 2015, 228-235; *Journal of Materials Chemistry A* 5, 2017, 6193-6199; *RSC Advances* 5, 2015, 33400-33406; *ACS Applied Materials & Interfaces* 8, 2016, 23425-23430), making the application of the negatively charged porous membrane a less innovative contribution. Additionally, the removal of zinc dendrites/accumulations is simply explained by the mutual repulsion of the negatively charged pore walls and zincate ions. In the reviewer's point of view, this explanation is of less scientific significance to related fields, as compared with any new understanding of the mechanism on zinc dendrite formation or methodologies to prevent zinc dendrite, which, however, is not provided in this work. Moreover, repulsion between the negative charges and zincate ions only occurs in a very short distance. Is this finding still applicable when there exists a gap between the negative electrode and membrane, which is adopted for zinc-bromine flow batteries in commercial demonstrations? In addition to the major comments mentioned above, there are several minor comments also need to be addressed.

(1) In line 72, the authors state that "if zinc dendrites form, they grow through the backward direction of the membrane". Evidence should be given to make such a claim.

(2) Given that carbon felt is good electrical conductor, why does zinc remain in the carbon felt due to disconnection after discharge using P0 membrane (line 163-165), but can be fully utilized within the carbon felt when P20 is used (line 213-214)?

(3) From the SEM images (Fig. S2), membranes (P15, 20 and 25) seem to present less pores than P0, which should lead to lower voltage efficiencies, but higher VEs are achieved in the battery tests (Table S1). Why? The reviewer is also curious about why the P15 membranes exhibit least pores.

(4) As shown in Fig. S3, P15 membrane presents a lowest $\text{Fe}(\text{CN})_6^{3-}$ permeability and therefore, it is expected to have a highest CE among the tested membranes. However, the highest CE is obtained by using P20 membrane. Explanation should be provided.

(5) The language needs to be further improved.

Response to Reviewer 1:

Comments to the Author

Dendrites is a long standing issue for batteries involving stripping/plating process, like lithium based batteries and zinc based batteries, which prevents the practical application of these batteries. This manuscript reported a dendrites free alkaline zinc-iron flow battery with long cycle life by employing a negatively charged nanoporous membrane. The innovative structure design in membrane, which performed significantly better chemical stability than traditional anion exchange membranes in alkaline medium, has produced an excellent battery performance even at a high current density of 160 mA cm⁻². This is a ground breaking work in the field of alkaline zinc based batteries from a well-respected group. I am sure the effort and novelty of this work are to be commended, which will have immediate and long-lasting impact for the zinc based batteries.

Therefore, I recommend the manuscript to be accepted and published in Nature Communications. Some minor comments are as following:

Response: Firstly, We really appreciate the reviewer's positive comments on our work.

1. In part of materials, the DS of SPEEK was calculated by ¹H-NMR, the calculation method including the NMR spectrum should be supplied.

Response: As suggested, the calculation method including the NMR spectrum were added in the revised version (Figure S1 in the revised version).

2. The permeability rate of Fe(CN)₆³⁻ should be provided.

Response: As suggested, the permeability rate of Fe(CN)₆³⁻ was calculated according to Fick's diffusion law and provided in the revised version.

3. From Figure 1, the SEM image of all the membranes demonstrate a sponge-like, symmetric structure, I just wondering why a porous membrane with asymmetric structure was not used.

Response: Indeed, the mechanical stability of a porous membrane with asymmetric structure is lower than does a porous membrane with symmetric structure. The efficiency of an alkaline zinc iron flow battery with an asymmetric structure porous membrane decreases faster than does a battery with a symmetric structure porous membrane as shown in Figure R1. Therefore, a porous membrane with symmetric structure was used in the work.

Figure R1. The performance of the alkaline zinc iron flow battery assembled with P0 membrane with different morphologies at the current density of 80 mA cm^{-2} .

4. For STEM measurement, why P20 and P0 were treated in a different way?

Response: For STEM measurement, the P20 and P0 were treated in a different way because of the fact that SPEEK in P20 was negatively charged, while PVP in P0 was positively charged. STEM measurement of P0 membrane stained with $[\text{PdCl}_4]^{2-}$ was carried out to confirm that P0 membrane was electrically neutral.

5. For better understanding the excellent performance of the battery with a negatively charged nanoporous membrane, I suggest moving Table S2 into the main article.

Response: As suggested, Table S2 was moved into the main article in the revised version (Table 1 in the revised version).

6. In Figure 5c, why the peak power density of the battery at 80% SOC was not given further?

Response: We appreciate the reviewer's attention to the details. The peak power density of the battery at 80% SOC was not measured since the current (the power density of the battery at 80% SOC at current density of 1080 mA cm^{-2} (9.72 A) was 1235 mW cm^{-2}) has run out of the full scale of our device (ArbinBT 2000, 5 V, 10 A).

Response to Reviewer 2:

Comments to the Author

This work reports a dendrite-free Zn deposition induced by a negatively charged nanoporous membrane. A high cycling stability of alkaline Zn/Fe battery is demonstrated. The authors claim that the rejection of zincate anion from the membrane due to the coulombic interaction.

The major concern on this work is that it does not provide a theoretical basis for this behavior. Why does the zincate rejection prevent dendrite formation? The current explanation is not enough. In addition, a direct evidence for zincate rejection is needed.

Response: First, we greatly appreciate the reviewer's constructive feedback on our work. Indeed, as mentioned in the manuscript, the zincate ions plating can be induced from the membrane direction to the 3D carbon felt framework direction through the mutual repulsion between the negatively charged zincate ions and the negatively charged surface and pore walls of the nanoporous membrane. Maybe we can not prevent dendrite formation, but we can control the the zinc dendrite to grow through the backward direction of the membrane, thus preventing the membrane from being broken up and avoiding the battery from short circuit. To further confirm our design philosophy and provide a direct evidence for zincate rejection, molecular dynamics simulations were carried to provide theoretical basis and support our conclusion. In addition, additional experiments were carried out in the revised version as well.

To confirm our design philosophy, an accelerated cycling experiment was performed by increasing the SOC of the battery to 85%. The higher SOC means longer charging-discharging time and more serious zinc dendrite, which would definitely accelerate the damage of the membrane caused by the zinc dendrite and further leading to the battery failure. And membranes with different content of of negative charge were fabricated and detected as well (P0, P15 and P25, with SPEEK content of 0, 15% and 25%). As a consequence, the capacity of the battery with a P0 membrane operated at 85% SOC decreased dramatically within 20 cycles (Figure R2), which showed much lower stability than does a battery with the same membrane operated at a low SOC (Figure 2d in the main article). While for the negatively charged nanoporous membranes,

the negatively charged sulfonic acid groups on the surface and pore walls could effectively repel negatively charged zincate ions diffusing to surface of the deposited zinc metal through the Donnan exclusion mechanism. The more negative charges on the surface and pore walls of the nanoporous membrane, the better cycling stability of a battery can be achieved. As a consequence, an alkaline zinc iron flow battery with a P25 membrane demonstrated a stable performance over more than 400 cycles even at a SOC of 85%, which is much more stable than does a battery with a P15 membrane at the same condition (Figure S10 in the revised version).

Figure R2. Accelerated cycling experiment of the alkaline zinc iron flow battery with SOC of 85% using P0, P15 and P25 membranes at the current density of 80 mA cm^{-2} .

At same time, simulations provided direct evidence for zincate rejection as well. Thus molecular dynamics (MD) simulation study was employed to calculate both Zn(OH)_4^{2-} rejection and ion transport through idealized membrane (Figure R3). Nonequilibrium molecular simulations (Video 2 (P20) in the attachments) clearly show that P20 with negative charges could reject Zn(OH)_4^{2-} anions, while allowing the permeability of Na^+ . This is also illustrated in the snapshots of the system using P20 membrane that has been taken at different times (Figure 4c in the revised version). By contrast, no significant rejection between P0 membrane and Zn(OH)_4^{2-} anions could be demonstrated in both

Video 1 (P0) (see the attachments) and snapshots (Figure 4c in the revised version). The distance between Zn(OH)_4^{2-} anions and P20 membrane (Figure 4e in the revised version) is much longer than the distance between Zn(OH)_4^{2-} anions and P0 membrane (Figure 4d in the revised version) because of the charge repulsion effect. The longer distance between Zn(OH)_4^{2-} anions and the P20 membrane means that the deposition of Zn(OH)_4^{2-} anions in membrane direction is much more difficult than those in backward direction of the membrane since the diffusion of Zn(OH)_4^{2-} anions into P20 membrane surface is prohibited, thus affording a very smooth zinc morphology (or zinc dendrite free, Figure 3 j-l in the main article). In comparison, from the simulations (Video 1 (P0) in the attachments, Figure 4 c and d in the revised version), it can be found that Zn(OH)_4^{2-} anions can be easily diffused into P0 membrane surface, which would definitely result in Zn(OH)_4^{2-} anions depositing in membrane direction, along with the formation of acrose type zinc dendrite (Figure 3 b and d in the main article).

Figure R3. Molecular dynamics simulation results using P0 and P20 membranes. a, Accessible solvent surface model of P0. b, Accessible solvent surface model of P20. The red balls represent sulfonic acid groups. c, Snapshots of the P0 and P20 membranes at different times, respectively. Snapshots are taken at 0 ns, 6 ns, 8 ns, and 10 ns, respectively. The blue balls represent Na^+ and the green balls represent $\text{Zn}(\text{OH})_4^{2-}$. d, Statistical distribution of Na^+ and $\text{Zn}(\text{OH})_4^{2-}$ at 10 ns using P0 membrane. e, Statistical distribution of Na^+ and $\text{Zn}(\text{OH})_4^{2-}$ at 10 ns using P20 membrane.

The other concern is that many factors affecting Zn deposition are neglected or uncontrolled. Pore size and distribution are not intensively characterized for the membranes. Also water uptake and wettability can be different. It should be clarified that the ions cannot permeate through the dense matrix for the membrane with high SPEEK content (In fuel cell community, this class dense membrane is used as a proton conducting membrane). These parameters can collectively influence Zn deposition behavior. Therefore, fine controls of the structures and properties are required to prove the surface charge-induced modulation of Zn deposition. It is widely accepted in Li metal electrode technology sector that more uniform and faster Li^+ flux through a porous membrane in contact with Li metal lead to a more uniform Li deposition. In this regard, not only the surface charge but also any difference in pore size, pore size distribution and electrolyte uptake can lead to a difference in Zn deposition behavior.

Response: The reviewer presents a very interesting point about the factors affecting zinc deposition. Even though the factors like pore size, pore size distribution and electrolyte uptake may lead to a difference in zinc deposition behavior, in-depth analysis has not been done in this manuscript as the main focus of this particular study was to utilize the negative charge property of nanoporous membrane figuring out zinc dendrite for alkaline zinc iron flow battery. Even this, the membrane with different PVP content (the PVP content in the polymer was kept as 40 wt.% to obtain a similar pore structure with nanoporous P0 membrane.) without charge was also detected as shown in the following figure. The results indicated that their cycle life (P0 membrane with the PVP content in the polymer was kept as 40 wt.% and 50 wt.%, respectively) is quite short as well, due to the zinc dendrite, which further confirmed the main role of the charge properties. In the meantime, it is very difficult to fabricate nanoporous membrane with identical structure (pore size, pore size distribution and electrolyte uptake) from cast solutions with different components.

Coulombic efficiency of the battery with a P0 membrane (the content of PVP in the cast solution was 40%) at 80 mA cm^{-2} .

However, as per reviewer's suggestion, we have characterized the pore size, pore size distribution and electrolyte uptake of the series membranes. The pore size and pore size distribution of the prepared membranes were analyzed by the Brunauer-Emmett-Teller surface area analyzer and all the prepared nanoporous membranes demonstrated a similar pore size distribution within a narrow range of 2 nm to 7 nm as shown in Figure R4.

Figure R4. N₂ sorption isotherms and pore size distribution curve of the prepared nanoporous membranes (determined by Barrett-Joyner-Halenda (BJH) analysis).

Figure R5 (Figure S5 in the revised version) exhibits the porosity and electrolyte uptakes of the prepared nanoporous membranes. The P0 membrane exhibited the highest porosity among the prepared nanoporous membranes, while the P15, P20 and P25 presented a similar porosity. Different from the tendency of porosity, the P15, P20 and P25 membranes exhibited an increased electrolyte uptake for both positive and negative electrolytes (Figure S5b in the revised version), whereas the electrolyte uptake of P0 membrane is in the range between the electrolyte uptake of P15 and P20 membrane. The increased electrolyte uptake was mainly attributed to the increased hydrophilic SPEEK content in the membrane. In-depth analysis of these results, taking P15 and P25, for instance, their porosity, pore size and pore size distribution are very close and there is not much difference for their electrolyte uptake (positive: 79% for P15 and 87% for P25; negative: 81% for P15 and 88% for P25).

However, an alkaline zinc iron flow battery using a P25 membrane demonstrated a stable performance over more than 400 cycles even at 85% SOC, which is much more stable than does a battery with a P15 membrane (~100 cycles) at the same condition (Figure S10 in the revised version) and surely much more stable than P0. Combining with molecular dynamics simulation results, it could be concluded that the improved cycling performance of the battery was mainly resulted from the negative charges on the surface and pore walls of the nanoporous membrane. On the other hand, currently the authors are not sure enough that whether or not the factors (pore size, pore size distribution and electrolyte uptake) affecting Li deposition apply to Zn deposition yet since their working principle and the state of the electrolyte (static or flow) are totally different. And as mentioned by the reviewer, it is widely accepted in Li metal electrode technology sector that more uniform and faster Li⁺ flux through a porous membrane in contact with Li metal lead to a more uniform Li deposition. This conclusion was obtained based on numerous in-depth studies of Li deposition. However, investigations of zinc deposition and zinc dendrite for alkaline zinc iron flow battery have been rarely reported, which makes it hard to draw a conclusion that the factors affecting Li deposition could apply to Zn deposition as well.

Figure R5. (a) Porosity and (b) electrolyte uptake of the prepared nanoporous membranes.

Indeed, we also tested the alkaline zinc iron flow battery using a dense P25 membrane at the current density of 40 mA cm^{-2} . But unfortunately, the battery can not be charged (Figure R6) due to the high area resistance of the membrane, indicating that the ions cannot permeate through the dense P25 membrane.

Figure R6. The charge curve of an alkaline zinc iron flow battery with a dense P25 membrane at the current density of 40 mA cm^{-2} .

The polarization and Zn deposition are directly related with area specific resistance and transference number, respectively, which can be experimentally quantified using impedance and junction potential measurement. Zincate diffusion, would be more important than other ion diffusions, was not investigated in this work.

Response: We greatly appreciate the reviewer's constructive comments on our work. As suggested by reviewer's comments, we have characterized the area specific resistance (Figure S8b in the revised version) and zincate diffusion (Figure S6b in the revised version) of the prepared nanoporous membranes as shown in Figure R7. The area specific resistance of the prepared membrane is well in accordance with the result of voltage efficiency of the battery (Figure S7 in the revised version). And all the membranes demonstrated extremely low permeability for $\text{Zn}(\text{OH})_4^{2-}$ anions as well, which is in accordance with the result of $\text{Fe}(\text{CN})_6^{3-}$ permeability.

Figure R7. EIS measurement (left) and the permeability of $\text{Zn}(\text{OH})_4^{2-}$ ions (right) through the prepared nanoporous membranes.

Indeed, the junction potential occurs when two solutions (different concentrations or different types) are in contact with each other. The transference number can be experimentally quantified using junction potential measurement according to the following equation:

$$E_j = (t_+ - t_-) \frac{RT}{F} \ln\left(\frac{a_1}{a_2}\right)$$

where E_j is the liquid junction potential of two solutions, t_+ and t_- are the transference number of the cations and anions, a_1 and a_2 are the activities of salt in the two solutions, R is the universal gas constant, T is the temperature and F is faraday's constant.

In a flow battery construction, the positive and negative electrolytes are separated by an ion conducting membrane, which means that there is no direct contact between the two electrolytes. On the other hand, for a flow battery, the positive and negative electrolytes are flow through the electrode, which is a dynamic process (the junction potential was

measured under a static condition). For these points, we are very sorry that we have not fully understood the junction potential for a flow battery and thus have no idea to figure out an effective method to measure the junction potential.

However, for better understanding the zinc deposition process, we carried out an rotating disk electrode (RDE) experiment. As shown in Figure R8, the zinc deposition is a diffusion controlled process. The limiting currents were plotted versus the rotation rate. The slope was fit using the Levich equation. According to the Levich equation, the diffusion coefficient of $\text{Zn}(\text{OH})_4^{2-}$ was $3.21 \times 10^{-6} \text{ cm}^2 \text{ s}^{-1}$.

Figure R8. (a) Rotating disk electrode measurements at rotating electrode speeds from 100 to 600 RPM using $0.2 \text{ mol L}^{-1} \text{ Zn}(\text{OH})_4^{2-}$ in 3 mol L^{-1} sodium hydroxide solution. (b) Linear relationships between the limiting current and the square root of the rotation velocity (Levich-plot).

The term 'backward Zn dendrite growth' is not fully verified. The observed behavior can be simply the progressive Zn deposition from the surface to the interior. If the authors stick to use the term, they may provide a theoretical basis for the backward dendrite growth. Electrolyte uptake, porosity, ionic conductivity, area specific resistance of the series of membrane should be measured for a clearer conclusion.

Response: We thank the reviewer for providing insightful comments to further improve the quality of our manuscript. As suggested, the electrolyte uptake (Figure S5b), porosity (Figure S5a), ionic conductivity (Figure S8a), area specific resistance (Figure S8b) of the series of membranes were measured and discussed in detail in the revised version. And a theoretical basis was provided in the revised version (Figure 4 in the revised version).

Response to Reviewer 3:

Comments to the Author

Comments on the paper entitled: “Negatively Charged Nanoporous Membranes: Toward Dendrites Free Alkaline Zinc Based Flow Battery with Long Cycle Life” by Zhizhang Yuan, Xiaoqi Liu, Yinqi Duan, Huamin Zhang and Xianfeng Li

This work reports on employing a negatively charged nanoporous membrane to eliminate zinc dendrites/accumulations in alkaline zinc based flow batteries. This issue is claimed to be well tackled by the repulsion effect of the negatively charged nanoporous membrane, thus resulting in a much improved performance of Zn-Fe flow batteries, in terms of cycling stability and power density. The results are interesting. However, the novelty is not high enough to meet the requirement of nature communications. First, porous membranes with and without charge on the surface of pore walls have been extensively studied for flow batteries (e.g., *Journal of Power Sources* 342, 2017, 327-334; *Journal of Power Sources* 353, 2017, 11-18; *Journal of Power Sources* 298, 2015, 228-235; *Journal of Materials Chemistry A* 5, 2017, 6193-6199; *RSC Advances* 5, 2015, 33400-33406; *ACS Applied Materials & Interfaces* 8, 2016, 23425-23430), making the application of the negatively charged porous membrane a less innovative contribution.

Response: First, we really appreciate the reviewer’s effort on our work and appreciate the reviewer’s attention to the applications of porous membranes for flow batteries. Regarding this comment, we would like to clarify below the innovative contribution, which we believe make this manuscript suitable to the high standard of Nature Communications.

Firstly, porous membranes with and without charges on the surface of pore walls have been extensively studied for flow batteries, especially for the vanadium flow battery (e.g., *Journal of Power Sources* 342, 2017, 327-334; *Journal of Power Sources* 353, 2017, 11-18; *Journal of Power Sources* 298, 2015, 228-235; *Journal of Materials Chemistry A* 5, 2017, 6193-6199; *RSC Advances* 5, 2015, 33400-33406; *ACS Applied Materials & Interfaces* 8, 2016, 23425-23430). And we have done many works on porous membranes for vanadium flow battery as well (e.g., *Chem. Soc. Rev.*, 2017, 46, 2199-2236; *Angew. Chem. Int. Ed.* 2016, 55, 3058-3062; *Energy Environ. Sci.*, 2016, 9, 441-447; *Energy Environ. Sci.*, 2016, 9, 2319-2325; *Energy Environ. Sci.*, 2013, 6, 776–781; *Energy Environ. Sci.*, 2011, 4, 1676–1679; *Energy Environ. Sci.*, 2012, 5, 6299–6303; *Adv. Funct. Mater.* 2016, 26, 210–218; *Adv. Funct. Mater.* 2015, 25, 2583–2589).

Currently we are invited to write a feature article for Chem. Commun. (DOI: 10.1039/c8cc03058h). In this feature article, we have summarized most of the ion conductive membranes for aqueous flow battery systems and found that currently no self-made nanoporous membranes have been employed and applied to the recently reported aqueous flow batteries, since it is not easy to fabricate nanoporous membrane with fine controls of the structures and properties that are just right for a given system with excellent battery performance. Therefore, to the best of our knowledge, this is the first time a self-made porous membrane was utilized in an alkaline zinc based flow batteries. And unlike traditional ion exchange membranes, the nanoporous membranes, isolating redox-active species from charge-balancing ions through pore size exclusion, have the characteristic of high chemical stability in both strong acid or alkaline and strongly oxidizing medium, which provides a new strategy toward the design and fabrication of high stability and performance membranes to substitute high cost Nafion series or unstable anion exchange membranes for alkaline zinc-based batteries.

Secondly, the key point of this work is the settlement of zinc dendrite/accumulation issue in alkaline zinc-based flow battery using the negative charge property of the nanoporous membrane. Zinc dendrite/accumulation is one of the most challenging problems for alkaline zinc-based flow batteries. Therefore, considerable efforts have been devoted to address the issue of zinc dendrite/accumulation in alkaline zinc-based batteries, of which introducing additives such as adding EtOH, Pb_3O_4 and Na_2WO_4 in the electrolytes is well known. Unfortunately, the additives normally result in a high polarization of the electrode, further leading to a decreased battery performance. Another effective way to address the zinc dendrite has been put forward through a backside-plating configuration that avoids short circuits from zinc metal dendrite in anode. However, this backside-plating configuration brings a two-fold increase in solution resistance over the frontside-plating configuration. In our work, by fine controls of membrane structures and properties, the alkaline zinc iron flow battery can afford excellent battery performance together with zinc dendrite/accumulation-free under current density varying from 80-160 mA cm^{-2} .

Lastly, due to the mutual repulsion between the negatively charged zincate ions in the alkaline medium and the negatively charged pore walls in the nanoporous membrane, the plating process of zincate ions was forced to take place mostly in the inner carbon felt direction, forming a smooth metallic zinc and leaving the membrane intactness. Given

this unique advantage, an 8 hours and a 7 hours plating/stripping process can be realized and excellent battery performance can be afforded at the current density of 40 mA cm⁻². Thus we believe that the data for the alkaline zinc iron flow battery with negatively charged nanoporous membrane is ample for such a novel design philosophy and is of high level among the recently reported aqueous flow battery systems.

Additionally, the removal of zinc dendrites/accumulations is simply explained by the mutual repulsion of the negatively charged pore walls and zincate ions. In the reviewer's point of view, this explanation is of less scientific significance to related fields, as compared with any new understanding of the mechanism on zinc dendrite formation or methodologies to prevent zinc dendrite, which, however, is not provided in this work.

Response: We thank the reviewer for providing valuable suggestions to further improve the quality of our manuscript. As we responded to Reviewer 2, simulations including molecular dynamics simulations and additional experiments were carried out in the revised version to provide a theoretical basis for our explanation. Please refers to the response to Reviewer 2 for more details.

Moreover, repulsion between the negative charges and zincate ions only occurs in a very short distance. Is this finding still applicable when there exists a gap between the negative electrode and membrane, which is adopted for zinc-bromine flow batteries in commercial demonstrations?

Response: We greatly appreciate the reviewer's constructive thought on the work. Unfortunately, we find that our design philosophy is not suitable for the battery, when a gap exists between the negative electrode and membrane, the zincate anions would deposit irregularly in the gap (Figure R9 right). On the other hand, the existing gap between the negative electrode and membrane can result in a sharply decreased battery performance (Figure R9 left). And we are sorry that this negatively charged nanoporous membranes cannot be adopted for zinc-bromine flow battery in principle, since the positively charged Zn²⁺ is employed as the active material. The negatively charged sulfonic acid groups in the nanoporous membrane could combine with the positively charged Zn²⁺, not exclude the positively charged Zn²⁺.

Figure R9. The battery performance when there exists a gap between the negative electrode and membrane (left) and the optical image of the deposited zinc metal on the electrode at the end of charging.

In addition to the major comments mentioned above, there are several minor comments also need to be addressed.

(1) In line 72, the authors state that “if zinc dendrites form, they grow through the backward direction of the membrane”. Evidence should be given to make such a claim.

Response: We thank the reviewer for providing insightful comments to further improve the quality of our manuscript. Indeed, as discussed in the manuscript, the the negatively charged zincate ions in the electrolyte were deposited irregularly, either in the membrane direction (Figure 3 a-d, Figure S11 a-c) or inner carbon felt electrode direction (Figure S12 a-c) when a battery assembled with a P0 membrane was charged. In contrast, when the same procedure was utilized to a negatively charged P20 membrane, the negatively charged zincate ions are mostly deposited in the inner carbon felt electrode direction, since the pore walls of a P20 membrane carried negatively charged sulfonated groups and thereby repelling zincate ions depositing in the membrane direction. As a result, a smooth membrane surfaces at the end of both charging (Figure S11 d-f) and discharging (Figure 3 m and n) were found, no zinc element on the membrane surfaces was detected by the EDS as well (Figure S11 f and Figure 3 o). In another word, if the zincate ions deposited in the membrane direction, the deposited zinc metal would pierce into the membrane, just as P0 membrane was. However, this assumption goes against our experimental results. Therefore, we speculate that if zinc dendrites form, they grow through the

backward direction of the membrane when the battery using a negatively charged P20 membrane.

To further confirm our design philosophy, we employed molecular dynamics (MD) simulation study (Figure R3) to calculate the rejection between the Zn(OH)_4^{2-} anions and the negatively charged P20 membrane (Figure 4 in the revised version). Nonequilibrium molecular simulations (Video 2 (P20) in the attachments) clearly show that P20 with negative charges could reject Zn(OH)_4^{2-} anions, while allowing the permeability of Na^+ . By contrast, no significant rejection between P0 membrane and Zn(OH)_4^{2-} anions could be observed in both Video 1 (P0) (see the attachments) and snapshots (Figure 4c in the revised version). Because of the charge repulsion effect, the distance between Zn(OH)_4^{2-} anions and P20 membrane (Figure 4e in the revised version) is much longer than the distance between the Zn(OH)_4^{2-} anions and P0 membrane (Figure 4d in the revised version). The longer distance between Zn(OH)_4^{2-} anions and the P20 membrane means that the deposition of Zn(OH)_4^{2-} anions in membrane direction is much more difficult (even is prohibited) than those in backward direction of the membrane since the diffusion of Zn(OH)_4^{2-} anions into P20 membrane surface is prohibited, thus affording a very smooth zinc morphology (or zinc dendrite free, Figure 3 j-l in the main article). In comparison, from the simulations (Video 1 (P0) in the attachments, Figure 4 c and d in the revised version), it can be found that Zn(OH)_4^{2-} anions can be easily diffused into P0 membrane surface, which would definitely result in Zn(OH)_4^{2-} anions depositing in membrane direction, along with the formation of acerose type zinc dendrite (Figure 3 b and d in the main article).

Figure R3. Molecular dynamics simulation results using P0 and P20 membranes. a, Accessible solvent surface model of P0. b, Accessible solvent surface model of P20. The red balls represent sulfonic acid groups. c, Snapshots of the P0 and P20 membranes at different times, respectively. Snapshots are taken at 0 ns, 6 ns, 8 ns, and 10 ns, respectively. The blue balls represent Na⁺ and the green balls represent Zn(OH)₄²⁻. d, Statistical distribution of Na⁺ and Zn(OH)₄²⁻ at 10 ns using P0 membrane. e, Statistical distribution of Na⁺ and Zn(OH)₄²⁻ at 10 ns using P20 membrane.

(2) Given that carbon felt is good electrical conductor, why does zinc remain in the carbon felt due to disconnection after discharge using P0 membrane (line 163-165), but can be fully utilized within the carbon felt when P20 is used (line 213-214)?

Response: The reviewer presents a very interesting point about the zinc accumulation. Shown in Figure S12 a-c is the optical and SEM image of the negative electrode of the

alkaline zinc iron flow battery assembled with a P0 membrane at the end of 65th discharging, where zinc accumulation in the carbon felt. When the battery was discharged, the successive zinc metal (dendrites) was stripped gradually and become disconnected, leaving portion still in the carbon felt, and other portion in the membrane (Scheme 1 and Figure 3 e-h in the revised version). However, it does not mean that the disconnected zinc metal resulted in the zinc accumulation in the carbon felt. Indeed, the The zinc metal (dendrites) pierced into the membrane (Scheme 1, Figure 3 e-h and Figure S11 a-c in the revised version) blocked the ion transport channel of the membrane, thus impeding the transportation of charge-balancing ions and further increasing membrane resistance. The increased membrane resistance thus result in zinc accumulation in the carbon felt when the battery was discharging (Figure S12 a-c in the revised version). And the zinc remained in the carbon felt (Figure S12 a-c in the revised version) is a long-term accumulation process.

While for P20 membrane, the negatively charged sulfonic acid groups on the membrane can prohibit zinc metal (dendrites) piercing into the membrane, leaving the membrane surfaces smooth (Scheme 1, Figure 3 m-o and Figure S11 d-f). The smooth membrane assure the ion conductivity of the membrane, allowing the free transfer of charge-balancing ions across the membrane without reducing the battery performance. On the other hand, the the negatively charged sulfonic acid groups on the membrane can force the zincate ions to deposit in the inner carbon felt electrode (Video 2 (P20) in the attachments), forming a dense and well conductive network between the deposited zinc metal and the carbon felt and affording a carbon felt/metallic zinc composite electrode. Therefore, during discharge process, the metallic zinc in the composite electrode can be fully utilized because of the dense and well conductive network between the metallic zinc and the carbon felt, thereby affording a nearly bare carbon felt electrode as ever without no obvious zinc accumulation (Figure S12 d-f).

(3) From the SEM images (Fig. S2), membranes (P15, 20 and 25) seem to present less pores than P0, which should lead to lower voltage efficiencies, but higher VEs are achieved in the battery tests (Table S1). Why? The reviewer is also curious about why the P15 membranes exhibit least pores.

Response: We appreciate the reviewer's attention to the details and insightful comments on the description of the results. Indeed, the voltage efficiency is closely related to the properties, e.g. ion conductivity, electrolyte uptake and area specific resistance. For the nanoporous membranes with negative charges, the charge property of the membranes

can adsorb more electrolyte and facilitate charge-balancing ions transport across the membrane (Energy Environ. Sci., 2013, 6, 776-781; Video 1 (P0), Video 2 (P20) in the attachments and Figure 4 c-d in the revised version), affording the membrane with higher ion conductivity (Figure S8a in the revised version), electrolyte uptake (Figure S5b in the revised version) and lower area specific resistance (Figure S8b in the revised version). As a consequence, the batteries with P20 or P25 membrane exhibited higher VE than does a battery with a P0 membrane. For P15 membrane, its ion conductivity, electrolyte uptake and area specific resistance is lower than does a P0 membrane, thus a lower VE is shown in the battery test.

For the formation of porous membrane pores, it can be explained by membrane formation mechanism, which has already been extensively investigated (Energy Environ. Sci., 2016, 9, 441-447; J. Power Sources, 2013, 233, 202-208; RSC Adv., 2014, 4, 40400–40406; RSC Adv., 2016, 6, 87104-87109; J. Membr. Sci., 1999, 155, 171-183; J. Membr. Sci., 1999, 156, 169-178; J. Appl. Polym. Sci., 1999, 74, 171-178; J. Membr. Sci., 1996, 117, 1-31; J.Mater.Chem., 2012, 22, 20512–20519). The membrane morphology is determined by the phase separation kinetics as well as thermodynamic equilibrium. Therefore, the morphology of a porous membrane can be effectively and conveniently tuned by adding a hydrophilic component to the cast solutions. The hydrophilic SPEEK in the cast solution can lead to an enhanced thermodynamic and a hindered kinetic hindrance during phase separation process (J. Membr. Sci., 1997, 138, 153-163), since it plays the role of reducing the miscibility of casting solutions with non-solvent as well as increasing the solution viscosity. Due to the excellent hydrophilicity of SPEEK, the solvent molecules can diffuse more easily into the solvent/non-solvent interface with higher SPEEK content in the cast solution, favoring the formation of porous membrane surface. As a result, the high content of SPEEK polymer in the cast solution results in an increased pore numbers in the membrane surface, as displayed in Figure S3 in the revised version.

(4) As shown in Fig. S3, P15 membrane presents a lowest $\text{Fe}(\text{CN})_6^{3-}$ permeability and therefore, it is expected to have a highest CE among the tested membranes. However, the highest CE is obtained by using P20 membrane. Explanation should be provided.

Response: The authors appreciate the reviewer's attention to the details and insightful comments on the description of the results. As proved by the permeability experiment, all the membranes demonstrated extremely low permeability for both $\text{Fe}(\text{CN})_6^{3-}$ and $\text{Zn}(\text{OH})_4^{2-}$ anions (Figure S6 in the revised version). Thus the battery with the prepared

membranes is expected to delivery high coulombic efficiency. To confirm the high coulombic efficiency of the battery, the battery performances were randomly selected and fluctuated. Due to the extremely low permeability of $\text{Fe}(\text{CN})_6^{3-}$ and $\text{Zn}(\text{OH})_4^{2-}$ anions, it is hard to discriminate the CE of the battery between high and low since the the value of the CE fluctuates over a very small range as shown in Figure R9 (the CE of the battery with the prepared membranes rose and fell as cycling proceeding, however, only fluctuating over a very small range). In the revised version, to avoid this confusion, we replace the table with a graph (Figure S7 in the revised version).

Figure R9. The CE of the batteries with P15, P20 and P25, respectively at the current density of 80 mA cm^{-2} .

(5) The language needs to be further improved.

Response: We appreciate the reviewer for pointing this out. In the revised version we tried our best to polish the English. Hopefully, it could meet the standard of Nature Communications this time.

Response to Reviewer:

Comments to the Author

The authors have adequately addressed the comments. I suggest that the paper be accepted for publication in the Journal.

Response: We really appreciate the reviewer's positive comments on our work.